# Do individual and institutional predictors of misconduct vary by country? Results of a matched-control analysis of problematic image duplications

**Daniele Fanelli**[1]*, **Matteo Schleicher**[1], **Ferric C. Fang**[2], **Arturo Casadevall**[3], **Elisabeth M. Bik**[4]

**1** Department of Methodology, London School of Economics and Political Science, London, United Kingdom, **2** Department of Laboratory Medicine and Pathology, University of Washington School of Medicine, Seattle, Washington, United States of America, **3** Department of Molecular Microbiology and Immunology, Johns Hopkins Bloomberg School of Public Health, Baltimore, MD, United States of America, **4** Harbers Bik LLC, Sunnyvale, CA, United States of America

* email@danielefanelli.com

**Data Availability Statement:** The anonymized data and R code used to generate all results are within the paper and its Supporting Information files. The

## Abstract

Pressures to publish, perverse incentives, financial interest and gender are amongst the most commonly discussed risk factors for scientific misconduct. However, evidence of their association with actual data fabrication and falsification is inconclusive. A recent case-controlled analysis of articles containing problematic image duplications suggested that country of affiliation of first and last authors is a significant predictor of scientific misconduct. The same analysis found null or negative associations with individual proxies of publication rate, impact and gender. The latter findings, in line with previous evidence, failed to support common hypotheses about the prevalence and causes of misconduct, but country-level effects may have confounded these results. Here we extend and complete previous results by comparing, via matched-controls analysis, articles from authors in the same country. We found that evidence for individual-level risk factors may be significant in some countries, and null or opposite in others. In particular, in countries where publications are rewarded with cash incentives, and especially China, the risk of problematic image duplication was higher for more productive, more frequently cited, earlier-career researchers working in lower-ranking institutions, in accordance with a "misaligned incentives" explanation for scientific misconduct. However, a null or opposite pattern was observed in all other countries, and especially the USA, UK and Canada, countries where concerns for misaligned incentives are commonly expressed. In line with previous results, we failed to observe a statistically significant association with industry funding and with gender. This is the first direct evidence of a link between publication performance and risk of misconduct and between university ranking and risk of misconduct. Commonly hypothesised individual risk factors for scientific misconduct, including career status and productivity, might be relevant in countries where cash-reward policies generate perverse incentives. In most scientifically active countries, however, where other incentives systems are in place, these patterns are not observed, and

data were anonymized to protect the identity of authors with image irregularities, in accordance with PLOS' data availability policy (https://journals.plos.org/plosone/s/data-availability).

**Funding:** The author(s) received no specific funding for this work.

**Competing interests:** "EMB regularly does consulting work on suspected science misconduct cases for publishers and research institutions, and gets speaker's fees to give talks about research integrity. She is an author on four uBiome (now Psomagen) patents (US20190050534A1 / US20180137239A1 / US20190078142A1 / US20200303070A1). EMB's company is indicated as an affiliation because she is an independent scholar, but the company did not provide resources that either directly or indirectly contributed to this study. DF advises and investigates allegations of misconduct for the Luxembourg Agency of Research Integrity, for which he receives regular honoraria. This does not alter our adherence to PLOS ONE policies on sharing data and materials."

other risk factors might be more relevant. Policies to prevent and correct scientific misconduct may need to be tailored to a countries' or institutions' specific context.

## Introduction

Understanding the psychological, sociological and structural factors that increase the risk of scientific misconduct is a core objective of research on research integrity and meta-science more generally. Identifying such factors is of theoretical interest, as it contributes to our understanding of causal patterns in human behaviour, as well as pragmatic interest, because it is the basis for designing policies and interventions to prevent research misconduct.

One of the most commonly discussed structural risk factors for scientific misconduct are "pressures to publish"–that is, unrealistic productivity standards imposed on academics, which may compel them to "cut corners" in order to sustain high levels of productivity and impact [1]. The "pressures to publish" hypothesis may be considered a special case of a more general "misaligned incentives" hypothesis, according to which scientists deviate from norms of good research practice to the extent that they have more to gain and less to lose from doing so [2]. Whilst the notion of pressures to publish and misaligned incentives is intuitive and plausible, evidence of its role as a causal factor for scientific misconduct is ambiguous. On the one hand, surveys support the widespread perception that academics are under pressure to publish, and that this may increase misconduct [3–5]. On the other hand, however, analyses of the literature have repeatedly failed to observe positive correlations between individual productivity or citation impact and poor research quality, at least when the latter was measured as the likelihood to correct or retract articles [6] or to publish over-estimated effect sizes [7,8].

The "funding effect" is another form of "misaligned incentive" very widely discussed and studied. A broad literature in clinical medicine suggests that industry sponsorship is more likely to report outcomes that are favourable to the sponsor, possibly due to a combination of methodological and reporting biases [9]. By extension, it seems plausible to hypothesize that privately funded research is at higher risk of scientific misconduct [10]. However, the opposite argument seems equally plausible: commercial interests linked to a research are a disincentive to research misconduct, because any short-term benefit accrued from misreporting research results is likely to be paid in the long term by financial losses caused by commercializing poorly performing products. Indeed, the nature and exact causes of the "funding effect" itself are still subject to some controversy [11].

A third factor that has attracted some research and controversy is gender. Statistics of the US Office of Research Integrity show that males are disproportionally represented amongst findings of scientific misconduct [12]. This could be explained by the fact that, on average, males are more aggressive, competitive and prone to taking risks (including with research misconduct) than females, or it could result from other structural factors and confounding variables, including the fact that men are more likely to hold multiple grants and may be less effective at negotiating their defence before and during an investigation [13]. Indirect support for the latter interpretation was offered by evidence that gender is not a significant predictor of retractions and corrections in the literature, or bias in meta-analysis [6,7].

A direct test of these and other causal hypotheses for scientific misconduct was conducted by some of us in [14], where we compared characteristics of publications that had been identified as containing or not containing problematic image duplications [15]. Using a matched-control analysis on a sample of such articles published in *PLoS ONE*, in [14] we observed

strong between-country differences, particularly for articles containing image manipulations that are more likely to be intentional (i.e. duplication categories 2 and 3, see below). Articles of which the first and/or last authors were based in China and India were at significantly higher risk of problematic duplications, but the variance was generally high amongst other countries, too.

Further, in [14] we found that the probability of image duplication was significantly higher for countries where researchers receive cash incentives for publishing in high-impact journals (in particular, China, Turkey, South Korea), and in countries that combine a developing economy with a hierarchical academic environment (as hypothesised by [16]). Conversely, the risk was significantly lower amongst countries in which publication incentives were in the form of career advancement (for eample, a tenure-track system), countries that combine developed (highly regulated) economies with a non-hierarchical academic environment, and countries that have national and legally binding policies against research misconduct. These patterns were interpreted as partially supporting a misaligned incentives explanation, in which cash rewards and low regulations and social control are significant risk factors for scientific misconduct [14].

Conversely, individual-level factors in the same analysis yielded a null or a negative support for the misaligned incentives and other common hypotheses, because the probability of image duplication was null or negatively associated with characteristics of first and last authors including gender, number of years of publication activity, average number of papers published per year, and average citation rates and journal impact factor. However, results of analyses at the individual-level were not deemed conclusive, because the strong between-country differences constituted a significant confounding factor [14]. Bibliometric data for authors in China and India, for example, were likely to be biased since these authors may publish part of their work in local journals not indexed by the ISI Web of Science database used in these analyses.

The present study aimed to resolve the apparent contradictions between country-level and individual-level patterns observed in [14], by completing and improving the analysis in three important ways. First, instead of matching articles by time of publication, we matched articles by country in which the first author operates, which allows to fully control any country-level confounding effect. Second, we focus our analysis on multivariable tests and include in all models an interaction term between career length of authors and their average publication rate per year. This interaction term tests the hypothesis that the career length of an individual modulates the relation between misconduct and publication rates, a hypothesis that [14] had not tested.

A third improvement from [14] consists in testing two commonly invoked risk factors:

- Funding source and employment status: we tested the funding effect on misconduct, by assessing the association between the probability of image duplication and the nature of funding acknowledged in each article or, as an alternative proxy, the kind of institution indicated as main affiliation for the first author.

- Institution ranking: to further probe the effects of cash incentives policies and pressures to publish, we assessed how the probability of image duplication varied with the institution's position in international university rankings. We hypothesised that the probability of image duplication might be higher for authors in lower ranking institutions. This hypothesis was particularly relevant to China, where high-ranking universities are known to reward authors with relatively modest cash amounts that are proportional to the actual citation impact of articles, whereas low-ranking universities give higher rewards based on the impact factor of the journal in which the article appears, which arguably encourages misconduct [17]. In particular, for each article published in the journal *PLoS ONE*, which is the source of our data,

"tier 3" institutions (which are the lowest ranking in the Chinese 3-tier academic system) offer on average 1,661 USD, which is over four times the average reward paid by "tier 1" institutions and over half the average salary of a newly-hired professor [18]. A publication in the top journals *Nature* and *Science* may yield, in some Chinese institutions, up to 165,000 USD dollars. First authors are typically the main or sole beneficiary of these cash rewards (70% of policies examined in [18] only award cash to the first author) and are therefore most likely to be influenced by them.

## Methods

### Sample of articles

We re-analysed a subset of the data used in [14], which in turn was a re-analysis of part of the data set produced by [15]. The latter consists in a large data set (N = 20,621) of articles whose images had been individually inspected for the presence of problematic image duplications. This sample was obtained by searching for articles with the keyword "Western blot" in 40 different journals and 14 publishers, and the images consist of molecular biology visual data, primarily photo compositions of Western Blots and other cell and tissue images. Image duplications were coded twice independently and classified in a scale of three categories, where, at one end, Category 1 are simple duplications that are most likely to result from simple error, and, at the other, Category 3 include misleading transformations that are most likely to be intentional (for further details see [15]). The sample used by [14] was a subset of this data, which was restricted to articles published in the journal *PLoS ONE* between 2013 and 2014, and it included N = 346 articles that were deemed to contain problematic image duplications and N = 868 controls that were deemed to contain none. All but one of these articles had two or more authors.

### Case-control matching

The matching was based on the country of the first institution appearing in the affiliation list in the article's WOS record, which is most likely to correspond to the first author's main or sole country of affiliation. We then matched each "case" (article that contained an image duplication) with two "controls", which were selected at random from the pool of articles that had the same first author's country of affiliation but had not been found to contain image duplications. Articles that lacked two exact country matches (N = 46) were excluded from the analysis.

Articles were not matched for any other characteristic. In particular, we did not match based on time of publication because the articles are all published in a relatively narrow time window of 2 years, which makes time an unlikely confounding factor.

This case-control matching was done blindly to any article characteristic. To avoid spurious inclusions, articles with the same first author (N = 2), and articles marked in the WOS as 'retracted publications' (N = 3) or 'corrections' (N = 2) were removed from the pool of potential controls. The final sample of papers therefore consisted of 293 papers with problematic image duplications and 2*293 = 586 controls (total N = 879).

### Data collection

Our analyses focused on characteristics of first-authors because first authors are, in biomedical research, typically the individuals most directly involved in all aspects of a research project. Further, previous analyses suggested that first authors exhibited patterns of particular interest

[14]. Furthermore, first authors are the main or sole beneficiary of cash publication rewards, at least in China [18]. For each treatment and control article, and for each first-author of these articles, we collected the following variables, either directly from records in the Web of Science (henceforth WOS) or from a bibliometric database that compiles data on individual authors using a name disambiguation algorithm [19]:

- *Team size*: number of co-authors listed in the article.

- *$10^*$ countries/author*: number of distinct countries listed as affiliations of the co-authors of the article, divided by the total number of authors. In the previous analysis, this variable was tested as the total number of countries [14], but we use here the ratio of countries to authors as it reduces the dependence of the two variables, allowing for a more robust test of the hypothesis that large international teams have a differential risk. This variable was then multiplied by ten, in order to improve the readability of the plots that show regression results. The regression estimates of main effects and interactions reported in all results and figures, therefore, represent rates of change associated with an increase in the ratio of countries to ten authors in an article. This transformation does not affect the statistical significance or any other aspect of results.

- *Years active*: number of years occurring between an author's first and last publication as recorded in the Web of Science (WOS), calculated as $year(lastpublication) - year(firstpublication) + 1$.

- *Papers/year*: for each first author in our sample, the total number of articles they co-authored, as recorded in the WOS, divided by the number of years they have been active (calculated as above).

- *Citations/paper*: total number of citations accrued by each first author, divided by the total number of articles they co-authored.

- *Female author*: gender of the first author, classified as female or male or unknown based on first name and country. Most names were classified by a commercial service (genderapi. com), and the records that remained unclassified were subsequently completed by hand, via direct inspection of information available online. In particular, Curricula Vtae and professional webpages were inspected for explicit gendering information and, if this was not available, based on photos, see [14] for details). This hand-coding was blind to the outcome (presence/absence of duplication in the paper). Names that remained unclassified (N = 51/879, 5.8%) were re-coded as "male". We used male as the default gender, based on the fact that most (i.e. 27) of these 51 unclassified names were Chinese or Taiwanese, and that about 75% of scientists in China are male [20]. The effect of this coding choice on results was checked with robustness tests reported in the Results section.

- *Cash incentives*: a binary category that identifies countries in which national or institutional policies offer cash rewards to authors who perform well in terms of publication number and/or impact. This is a recoding of the variable used in [14], which was based on a classification by [21]. In practice, in the present sample, this variable separates China and South Korea from all other countries.

- *German-developmental*: a binary category that identifies countries that, according to an independent sociological and historical analysis, exert the least effective social control due to a combination of hierarchical academic culture and low regulatory standards [16]. This variable is a recoding of the analogous variable used in [14], which had distinguished three levels

of social control and had excluded all non-classifiable countries. Here the variable is recoded as binary, to separate China, Japan and Korea from all other countries.

In addition to the variables above, all of which had been tested previously, the following new variables were recorded.

- *University rank* and *non-ranked institutions*: rank of each first author's affiliated university in the 2017 Shanghai Academic Ranking of World Universities. The Shanghai ranking was chosen because it includes a greater number of Asian institutions, and the 2017 version because it includes a larger list of institutions compared to previous versions (N = 800 vs. N = 500). Each institution has a numerical score between 0 and 800, but only the top 100 universities have a unique numerical score, whereas lower ranks are classified in percentile tiers: 101–150, 151–200 etc. We recoded the Shanghai score in eight tiers, from tier 0 (1–100) to tier 7 (701–800). Affiliations with universities that were not included in the Shanghai ranking were given a score of 8. Affiliations with non-university institutions were given a score of 0, and marked out with a separate binary variable ("non-ranked inst."). Therefore, regression estimates on institution rank reflect the increase in the odds of image duplication as the position in the Shanghai ranking decreases (its rank score increases by one step). Regression estimates for the non-ranked institution variable reflect the difference between the odds of image duplication for these non-ranked institutions and those for the top-ranking university institutions. When an author indicated multiple affiliations, we recorded the first (and therefore presumably main) listed affiliation. In addition to the Shanghai ranking, which has global application, we classified Chinese universities according to the national 3-tier system [18]. We used publically available lists of "tier 1" [22] and "tier 2" [23] universities, and classified the remaining as "tier 3".

- *Industry/mixed funding*: the funding sources listed in each articles' WOS record were retrieved and categorized as either "industry", "public", "non-profit research organisation", "university" and "mixed" by MJS. Since most records fell either in the "public" category or in the "mixed" category, the variable was recoded as a binary variable, separating an "industry/mixed fund" category from purely publically funded (i.e. university, non-profit etc.) research. As an alternative proxy for sponsorship influence, we recorded the nature of the affiliation listed in the affiliation section. Since in WOS records affiliations are listed in the same order as the authors, and since presumably authors list their multiple affiliations in decreasing order of importance, the first affiliation listed is likely to be the most representative for the first author.

- *Type of affiliation*: this was classified by MJS as "university", "industry", and "hospital" (total sample size, respectively, N = 742, N = 98, N = 39). Since the latter two categories were sparsely represented, they were merged together. Note that this variable overlaps with the previous two (hospitals and industries are non-ranked institutions, and authors affiliated with them may also be more likely to receive industry funding).

## Analyses

All main results were derived by running a multivariable conditional logistic regression model of the form:

$$logit(\pi) = log\left(\frac{\pi}{1-\pi}\right) = \alpha + \sum_{s=2}^{h} \alpha_s X_s + \sum_{i=1}^{k} \beta_i X_i$$

In which $X_s$ are the strata (triplets of case and controls), and $X_i$ are the explanatory variables and interactions included in the model. In practice, the strata component of the regression equation is conditioned out of the maximum likelihood estimation, and the resulting analysis can be succinctly described by the following model:

$$log\left(\frac{P_{dup}}{1 - P_{dup}}\right)$$
$$= \alpha + \beta_{team}X_{team} + \beta_{10count/aut}X_{10count/aut}$$
$$+ \beta_{team*10count/aut}X_{team*10count/aut} + \beta_{years}X_{years} + \beta_{papers/y}X_{papers/y}$$
$$+ \beta_{years*papers/y}X_{years*papers/y} + \beta_{cites/paper}X_{cites/paper} + \beta_{female}X_{female}$$
$$+ \beta_{unirank}X_{unirank} + \beta_{no-rank}X_{no-rank} + \beta_{industry+mixed}X_{industry+mixed}$$

where $P_{dup}$ is the probability of observing an image duplication in one of the three articles within each stratum, therefore in relative terms. The coefficients for the non-interacted variables (*citations/paper*, *female*, *institution rank*, *non-ranked* and *industry/mixed funding*) express the change of the log odds (or the odds ratio, if exponentiated) corresponding to a one-unit increase in the value of the variable whilst holding all other variables constant (note that some variables were re-scaled to make the plots clearer, so their coefficients reflect the new scale, as indicated above). The coefficients of the interaction terms (i.e. *team size\*countries/author* and *years active\*papers/year*) measure the rate at which the coefficient of either one of the variables changes as the value of the other variable changes. Therefore, the coefficients for the corresponding non-interacted variables (i.e. *team size*, *countries/author*, *years active*, *papers/year*) correspond to the change in log odds associated with a one-unit increase for that variable when the other variable in the interaction has value zero and all other variables are held constant.

This project was reviewed and approved by the Research Ethics Committee of the London School of Economics and Political Science.

## Results

### Descriptive statistics

Of the 293 papers with problematic images that could be matched with two controls and therefore included in the analysis, N = 72 were of duplication Category 1 (most likely to reflect unintentional error), N = 155 of Category 2, and N = 66 of Category 3 (most likely to represent intentional misrepresentation). Most articles had a first author based in China (N = 136) or USA (N = 59), followed at much distance by Italy (N = 11), South Korea (N = 10), Canada (N = 9), France (N = 7), India (N = 7), the UK (N = 7), and Germany (N = 5), whereas the remaining 19 countries had N≤4.

### Risk of category 2 and 3 duplications by country groups

If, following [14], we limited the sample to duplication categories 2 and 3 (most likely to reflect intentional manipulation) and we tested the explanatory factors on a sample with all countries combined, we observed no significant relation with the odds of image duplication, except for a weak positive association for institution rank (Fig 1A). If we compared, again following [14], Western/industrialised countries to the others, the positive association with institution rank was only observed in the latter group, whereas the former showed a significant interaction between years of activity and productivity (Fig 1B and 1C).

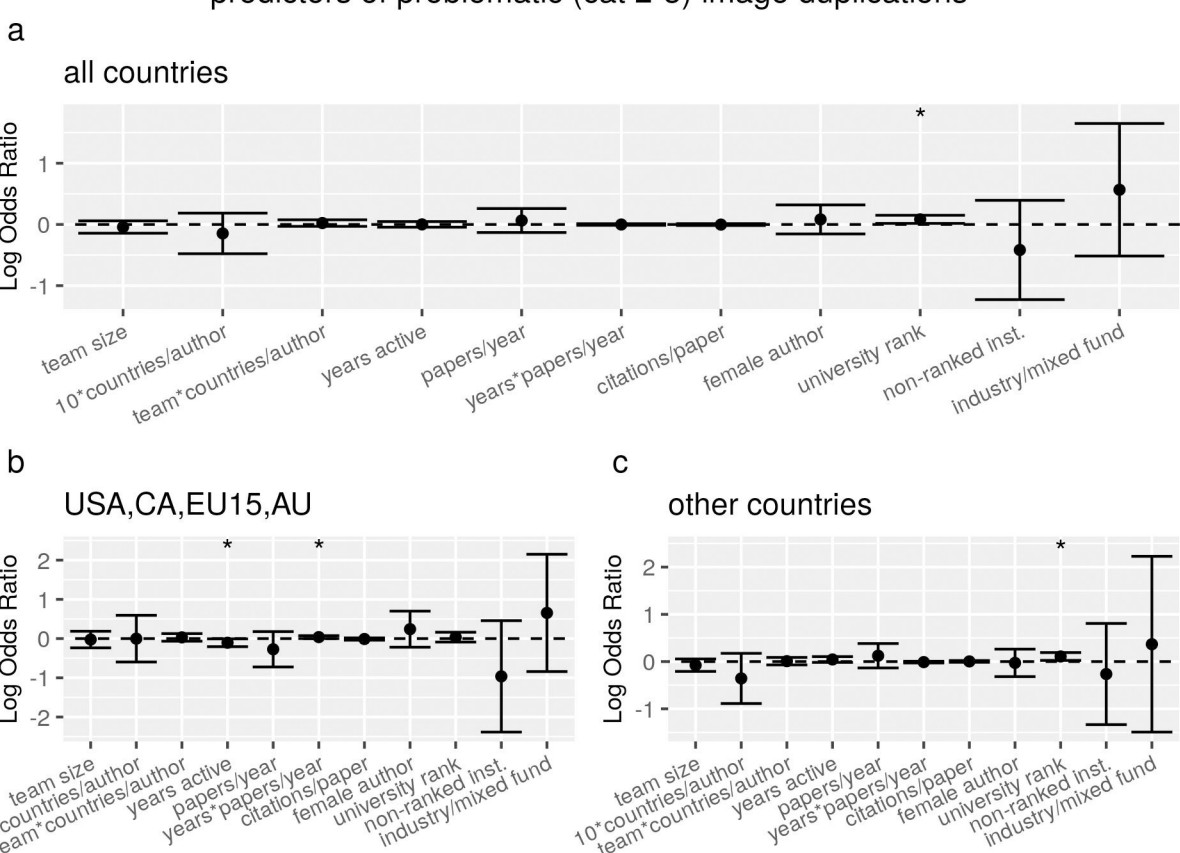

**Fig 1.** Regression estimates of the association between characteristics of articles (first authors) and the odds of containing a problematic (category 2 or 3) image duplication, for all articles (a) or country subsets (b, c). Each panel shows results of a separate conditional multiple regression analysis, run on a matched-control sample in which each paper with image duplications (N = 293) was matched with two controls with the same country of affiliation of first author. Error bars represent 95% confidence intervals, and the dots and stars above error bars indicate rejection of the null hypothesis of no association (i.e. OR = 1, dotted line) at conventional levels of statistical significance (˚P<0.1, *P<0.05, **P<0.01, ***P<0.001).

Marked differences were observed if the same regression model was run on subsets of countries that had theoretically relevant differences, which previous analyses showed to be significantly associated with problematic image duplications [14]. In particular, papers with first authors from the two countries within our sample that have a cash-based incentives policy (that is, CN and KR) exhibit positive associations with institution rank score, career length, papers/year and with their interaction, which are not observed at all in the other countries (Fig 2A and 2B).

If countries were divided by another variable that [14] had identified as significant, i.e. the socio-culturally relevant distinction of German-developmental countries vs. other, the effects of career length and productivity were no longer significant or significantly different between the two groups (Fig 2C and 2D).

If the analysis was limited to China on the one hand and USA, UK and Canada on the other (to increase sample size, we tested these three countries together under the assumption that they are relatively homogenous socio-culturally), the profile difference appeared even more extreme. USA, CA and UK only exhibited a weak (not formally significant at the 0.05 level) negative association with citation rates, and null associations with the other factors, most of which were in the opposite direction relative to the China subset (Fig 2E and 2F).

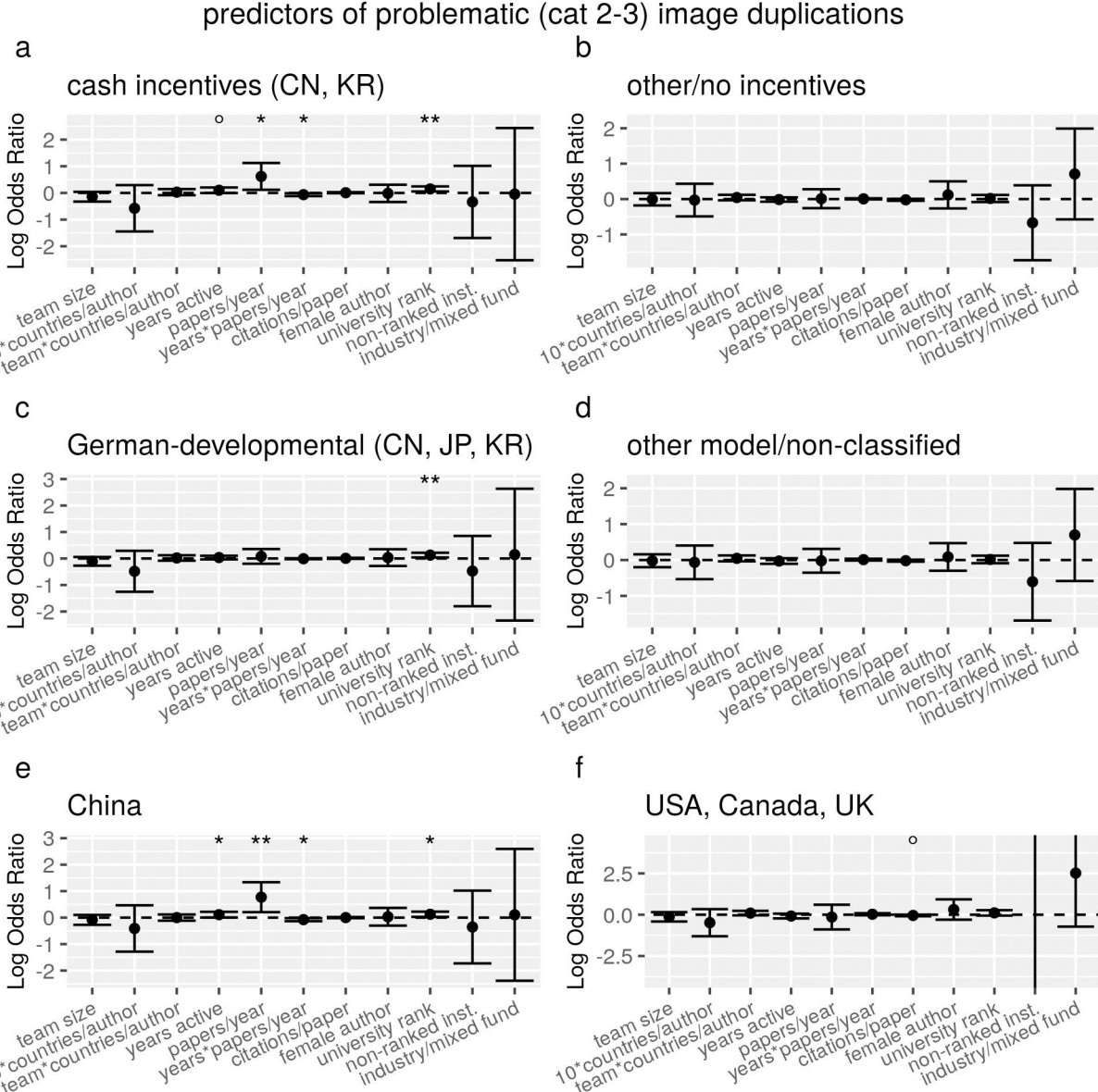

**Fig 2. Regression estimates of the association between characteristics of articles and of first authors and the odds of containing a problematic (category 2 or 3) image duplication, for all articles grouped according to theoretically relevant characteristics of the country of first authors.** Each panel shows results of a separate conditional multiple regression analysis, run on a matched-control sample in which each paper with image duplications (N = 293) was matched with two controls with the same country of affiliation of first author. Error bars represent 95% confidence intervals, and the dots and stars above error bars indicate rejection of the null hypothesis of no association (i.e. OR = 1, dotted line) at conventional levels of statistical significance (°P<0.1, *P<0.05, **P<0.01, ***P<0.001).

The main effects and interactions of team size and number of countries are weak and non-statistically significant in all analyses, but they exhibit effects that are in the same direction observed in [14] (where the total number of countries was used instead of the per-capita number) and show differences between all pairs of subgroups (Fig 2A–2F).

## Interpretation of findings

To help interpret the findings in Figs 2 and 3 reports probabilities calculated from the models reported in Fig 2A and 2B. We fixed the values of gender and funding status, and assessed how

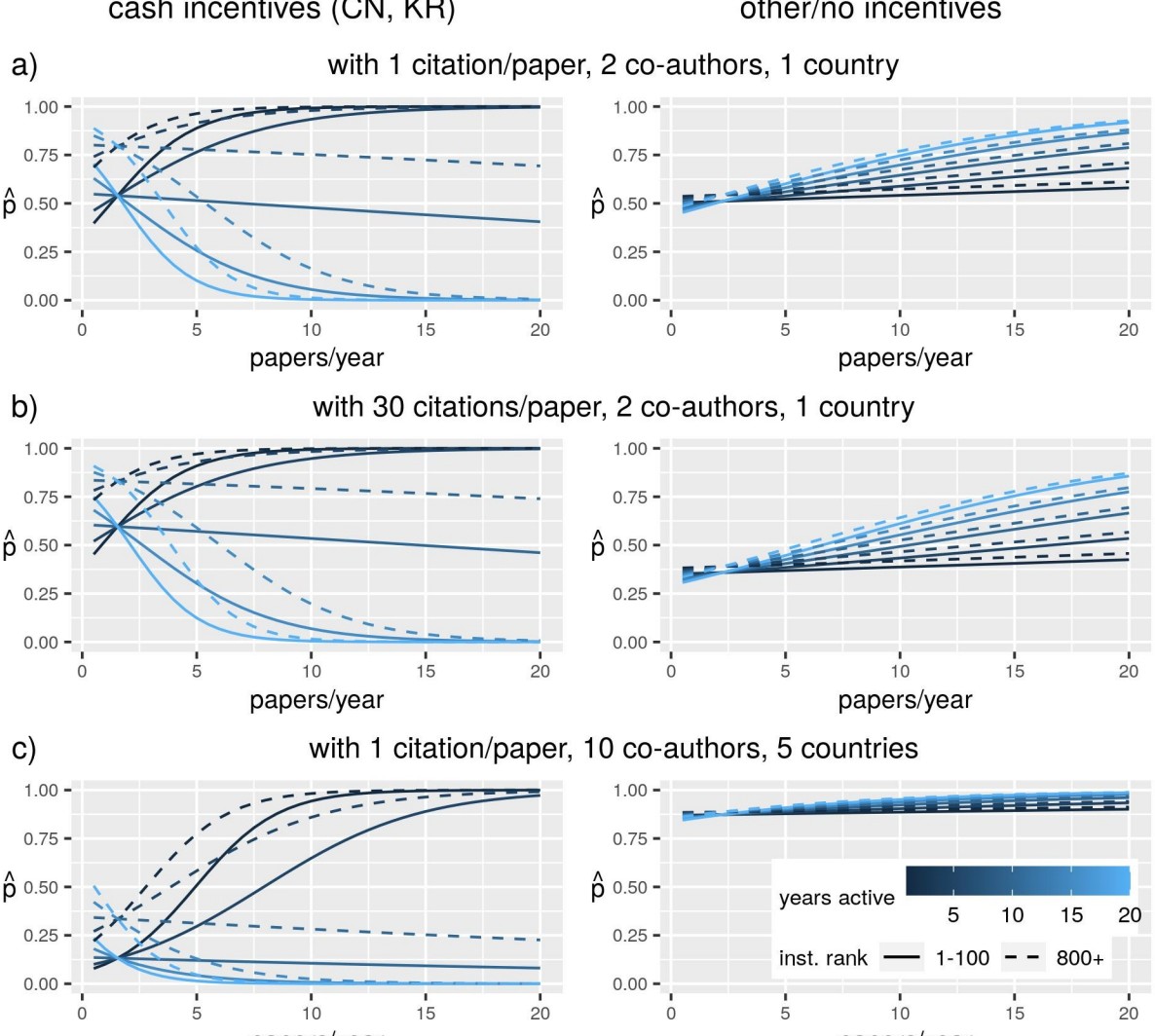

**Fig 3. Fitted probabilities calculated from the regression results shown in Fig 2A and 2B.** Each panel shows the rate of change of the relative probability of being a "treatment" (article with problematic category 2 or 3 image duplication) rather than a "control", as the values of some author and team characteristics are held constant (those indicated in the subtitle) whilst other characteristics vary (indicated in the figure or in the panel titles).

the estimated relative probability to have a category 2–3 duplication (note: the probability for treatment vs. controls in our sample, not in the general population) changed as we varied the average productivity of the author, their career length, the ranking of their institution, their average citation rate, the number of co-authors in the article and the number of countries. The resulting fitted probabilities may not represent any real combination of characteristics and therefore their value should not be taken literally, but rather in relative terms, i.e. as an illustration of how the risk of image duplication varies with varying conditions, according to our model's estimates.

Amongst cash-incentives countries, the relative probability of reporting a problematic duplication increases as productivity increases for individuals with shorter careers, and decreases for those with longer careers, regardless of other conditions (Fig 3, left panels). The overall probability is higher if the author works in a low-ranking institution and if the author is highly cited, whereas it decreases when the paper was co-authored with a larger, international team (Fig 3B and 3C, left panels).

The opposite is observed in the other countries. There is no relevant association between the risk of problematic duplication and any combination of productivity, career length and university ranking (Fig 3, right panels). However, the risk is substantially lower for authors who are highly cited (Fig 3B), and it is higher for large international teams (Fig 3C). Note how these effects are the exact opposite of what is observed in the two cash-incentives countries.

## Robustness of findings and secondary analyses

The main variables and analyses presented above had been pre-registered (osf.io/uzykd) as part of a plan to repeat all of the analyses in [14], testing each variable in univariate tests and then running multiple regressions as a secondary analysis. In this article we focused on the multivariable analysis, mainly for brevity and because multivariable analyses constitute more accurate tests of statistical associations. Data and code included with this article can be used to run all univariate analyses.

The multivariable model used in this study is analogous to that presented in [14], except for the recoding of number of countries into countries per author and for the inclusion of an interaction term between career length and average publication rate—variables that in [14] were treated as independent effects only. In retrospect, an interaction term should have been included in the previous analysis. If the recoded country/authors variable and interaction are included in a re-analysis of data in [14], similar results to this analysis are obtained: both the interaction terms of *team size* * *countries/author* and *career length* * *papers/year* yield significant signals, but only in cash-incentives countries (respectively, OR = 5.146, z = 2.158, P = 0.031 and OR = 0.122, z = 1.954, P = 0.051. Full output of these analyses are included in the R code, see SI).

We explored the possible presence of other interaction effects between career length and ranking or citations, but we observed no significant interaction effect and/or notable difference between cash-incentives and non-cash incentives, suggesting that the patterns reported in Figs 2 and 3 are specific to this categorization.

If a binary variable representing type of affiliation (i.e. university or hospital affiliation vs. other kind of affiliation) was added to the model in place of the "non-ranked institution" and the "industry/mixed funding" variables, results were similar, showing a lower probability of duplication for non-university affiliation in the non-cash incentives countries (b = -0.76±0.45, z = -1.78, P = 0.0759.).

Adding a categorical variable that reflects the three "tiers" recognized amongst Chinese universities (total sample size for Tier 1, 2 and 3, respectively N = 175, N = 62 and N = 170) did not change the results nor did it significantly increase the fit of the model ($\chi^2$ = 1.1092, df = 2, P = 0.574). Relative to tier 1, tier 2 and tier 3 university affiliation showed a positive but weak (not statistically significant) association with problematic image duplications (b = 0.36±1.43, z = 1.054, P = 0.292 and b = 0.21±1.23, z = 0.784, P = 0.433, respectively), suggesting that the misaligned incentives issue is less directly connected to the Chinese national tier system, and more directly linked to international rankings.

If gender was recoded attributing all unclassified individuals as "female" instead of "male", results were identical, showing very small and non-statistically significant effects of gender in

both cash- and non-cash incentives countries (respectively, b = -0.06±0.94, z = -0.328, P = 0.7430, and b = 0.16±1.17, z = 0.785, P = 0.432).

In a previous encoding of the "industry and mixed" dummy variable, which incorrectly included all types of mixed combinations of funding, we observed a significantly lower probability of duplication for this category amongst non-cash incentives countries. We explored this possibility further, by testing, in the same multiple regression model, all dichotomizations of funding sources, i.e. dummy variables for "no-profit/mixed fund", "university/mixed fund", "government/mixed fund" (see S1 Code). All of these dummy variables tended to be negatively associated with the probability of category 2–3 duplications in non-cash-incentives countries, but the null hypothesis was far from being rejected in all cases. Therefore, we found no evidence for systematic effects of funding of any types, once controlling for the other factors discussed in this analysis.

## Risk of category 1 vs. category 2,3 duplications

All results presented above are based on duplications of category 2 and 3, because these are more likely to represent intentional misrepresentations. In agreement with this interpretation, [14] noted that most univariable statistically significant associations with theoretically relevant variables occurred amongst category 2 and 3 duplications. However, the frequency of duplication types is unequally distributed amongst countries. In particular, the probability of producing duplications of more problematic kinds is lower in the USA and higher in China, with other countries having intermediate frequencies (Fig 4). Indeed, a logistic regression model suggests that the risk of producing category 2–3 rather than category 1 duplications is, holding all other variables constant, significantly lower for the USA relative to other countries, whereas that of China is as high or higher (Fig 5). Further, holding country and all other variables constant, the risk is significantly lower for highly-cited authors (Fig 5).

If the same set of variables is fitted with an ordinal model, to assess the risk of category 1 vs. 2 vs. 3, similar results are obtained overall, but the threshold separating categories 2 and 3 is non-significant, which corroborates previous suggestions that category 2 and 3 do indeed represent a distinctive and relatively homogeneous category of problematic image duplications compared to category 1 [14].

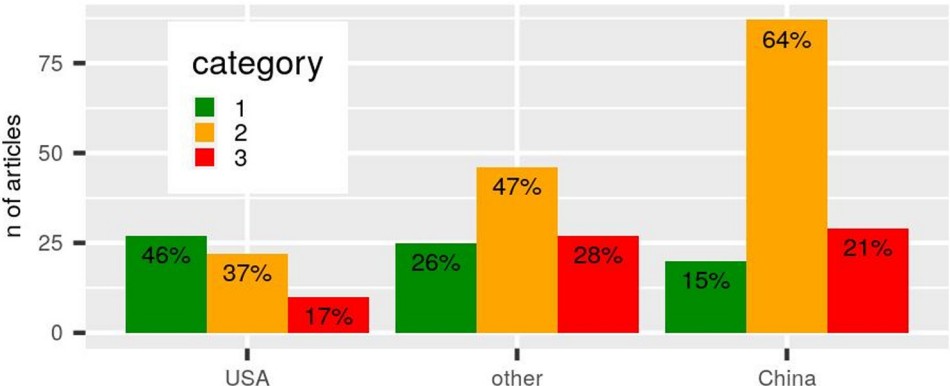

**Fig 4. Number and percentage of problematic image duplications in our sample, partitioned by country of first author and by severity of duplication (where category 3 is the most problematic, because it is most likely to reflect intentional misrepresentation of data).**

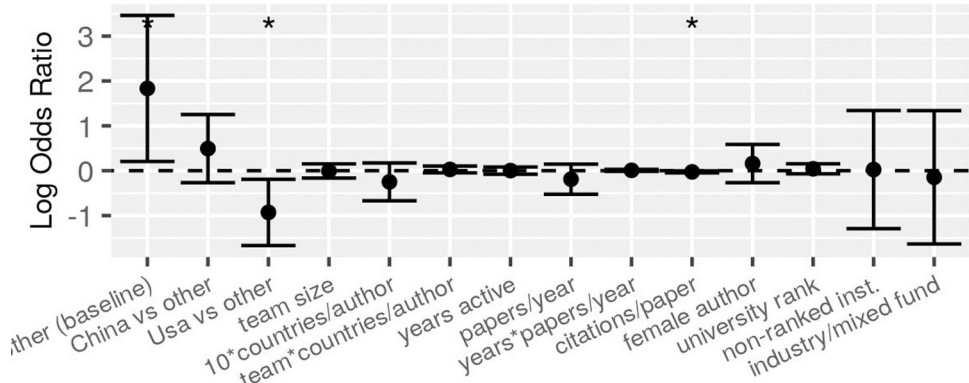

**Fig 5. Binary logistic regression estimates of the association between characteristics of "treatment" articles in our sample (N = 293) and the odds of reporting an image duplication most likely to reflect intentional misrepresentation of results (i.e. category 2 or 3).** Error bars represent 95% confidence intervals, and the dots and stars above error bars indicate rejection of the null hypothesis of no association (i.e. OR = 1, dotted line) at conventional levels of statistical significance (˚P<0.1, *P<0.05, **P<0.01, ***P<0.001).

## Discussion

A previous analysis of the same data set had identified country of author affiliation as a highly significant modulating factor for the risk of scientific misconduct [14]. This study completed and extended that analysis by assessing how the probability of duplicating images varies with commonly hypothesised risk factors for scientific misconduct, once country-level effects are controlled for. Results of this and the previous analysis suggest that some common hypotheses about the social and psychological determinants of scientific misconduct are not or negatively supported, whereas other hypotheses may or may not be supported, depending on the country of affiliation of the author, possibly due to the structure of incentives policies that operates in the country.

The hypothesis that gender is a risk factor for misconduct is not supported. In none of our analyses was there any significantly higher risk for men or women. In most analyses the estimated risk was very close to equal (Figs 1 and 2). Precision in gender classification tends to be lower for East Asian countries, and a small percentage of names was not classifiable and coded as male (5.8%). However, classifying these names as female did not alter the results. More importantly, the lack of difference between genders was observed across Western European and North-American countries, where names are typically gendered and therefore errors in gender attribution are extremely low. Along with similar evidence obtained when comparing retractions and corrections [6], and bias in meta-analyses [7], these results do not provide any support for the hypothesis that psychological differences between men and women modulate the risk of misconduct. The over-representation of males in US ORI findings of scientific misconduct remains to be explained, but it may be due to confounding factors such as gender differences in the number of grants held, the number of publications produced, or other traits that affect the likelihood that an ORI investigation is initiated and concludes with a finding of misconduct, as suggested in [13].

The hypothesis that private sponsorship may increase the risk of misconduct was inconclusively tested, because our test had very low statistical power. In all analyses we failed to observe the predicted association between risk of duplication and private funding. However, the number of studies in our sample that were fully or partially funded by industry was very small

(N = 22 in total, and N = 14 for category 2–3), which severely limited our ability to detect any pattern. Exploratory analyses of other funding categories, in which the numbers were more balanced, also failed to suggest any systematic pattern. Therefore, the possibility of a link between funding source and risk of misconduct, whilst not especially suggested by our findings, remains to be assessed, and our findings do not challenge the established association between funding sources and bias [9]. These observations are consistent with earlier studies that operated a distinction between research bias and fraud [24].

Evidence for the other hypotheses is more complex than commonly presumed. In China and Korea, countries in which high-impact publications are rewarded financially, the highest probabilities of manipulating images are observed in articles with authors who are highly productive, early-career researchers, have high citation rates and work in small, local collaborations and in low-ranking institutions (Fig 3). Further, authors from these countries appear generally to be at a higher risk of misconduct (Fig 5), as already suggested in previous analyses [6,14]. However, radically different patterns were observed in the other countries. Here the estimated risk is lower in small local collaborations, regardless of an author's productivity, career length or institution's ranking. And being a highly cited author is associated, in these countries, with a significantly lower risk of problematic duplications (Fig 3).

Most of the differential effects observed in China and Korea relative to other countries may be plausibly explained as resulting directly or indirectly from the incentives created by a cash-reward policy for high-impact publications. This hypothesis is statistically supported by our results, in which the "cash incentives" country category was associated with significantly more effects than alternative country categorizations (Fig 2). The hypothesis also offers the most coherent explanation of our multiple findings. Cash rewards may encourage authors to "cut corners" in their work, particularly if such authors work in low-ranking institutions, where salaries are low and cash rewards are largest [18], and if they are early-career researchers, and therefore have less to lose and more to gain from taking risks [25]. Working in a small and local team (a team from a low-ranking institution) may also reduce the level of mutual control and criticism, increasing the risk. Further, since cash incentives are typically only given for high-impact publications in international journals, these problematic authors would be expected to publish more articles in English-speaking journals and seek high levels of citation, which would result in the observed positive correlation between productivity, citation rates and risk of image duplication. Conversely, authors in countries with different types of policies have none of these incentives, and instead are likely to see their work scrutinised in proportion to its impact, which would plausibly result in highly cited authors being less likely to produce papers with manipulated results. The risk of image duplications might increase in larger international collaborations if these collaborations are more likely to include co-authors from high-risk countries and conditions.

Fully or partially alternative explanations, however, are also possible. For example, we cannot exclude that researchers in USA, UK, Canada and other countries have greater awareness of policies and principles of research integrity and have also access to better resources, which might enable them to either avoid forms of misconduct or, more cynically, to misrepresent data in subtler forms that are harder to discover (for example, they could be manipulating images in subtler and more elusive ways). Other entirely alternative explanations might invoke, for example, general socio-cultural differences between the countries involved. One of the referees on this manuscript, in particular, suggested that countries like China tend to be guided morally by a "culture of shame", in which individuals strive to meet social expectations by any means necessary, and shame arises primarily from being caught. This is in contrast to other countries, including Western, Judeo-Christian countries, which may be dominated by a "culture of guilt", in which the moral unit is the individual, not the collective. A "shame

culture" vs. "guilt culture" variable could represent an alternative explanation of at least some of the differential patterns of misconduct observed across countries. Whilst plausible, all of these explanations are merely speculative, and may be tested in future studies.

What can these results, combined with previous analyses, tell us about the controversial pressures to publish hypothesis? In our interpretation, they do not support it, but they do offer partial support of the hypothesis that misaligned incentives may increase the risk of misconduct. The pressure to publish hypothesis is not supported because the only significant patterns we detected in these and previous analyses support a role for cash incentive policies, which do not constitute real "pressures" on researchers, but rather an incentive that is misaligned with the scientific norms of disinterestedness [26]. Researchers in cash-incentives countries may of course feel additional kinds of institutional pressures, analogous to those reported by scientists in other countries [4,27]. However, most scientifically active countries do not exhibit the patterns predicted by the pressures to publish hypothesis, including countries like the UK or the USA, where genuine "pressures to publish" may be imposed on researchers in the form of bibliometric performance evaluation standards. To any extent that this and previous analyses reflect general patterns of research misconduct, current evidence suggests that in the latter countries, incentives are operating as they should: the higher the impact an author accrues, the less likely they are to be authors of flawed or falsified findings, regardless of their productivity, career level, or institution's reputation. Further, in these countries, privately sponsored research appears to be at a similar or lower risk of producing fraudulent results.

Our results should not be interpreted to mean that the *perception* of pressures to publish is not relevant in *individual* cases of scientific misconduct. To the contrary, the history of individual cases of misconduct often suggests that stress and performance expectations are contributing factors [28,29]. However, findings in this and multiple previous analyses fail to support the claim that pressures to publish are a systemic and generalized risk factor for misconduct, and they suggest that the relation between institutional incentives and research quality is complex and requires nuanced and contextualised analyses.

Before drawing conclusions, however, a few important limitations of our analyses must be emphasised. First, our data set is not representative of all forms of data manipulations and all research areas. We analysed a single form of data manipulation (and/or problematic error), measured in articles that were all published in a single journal. Further, the studies were selected for screening based on their use of relatively similar types of methodologies (i.e. results in the form of cell or gel images), and thus represent a rather narrow range of research types. Therefore, incentives and risk factors that we found to be significant in this study may not apply to other forms of misconduct and other journals and research areas. Furthermore, the study was not designed to sample different countries and, because of the matching procedure, our data were dominated by authors from China and USA. Therefore, our results may not be generalizable to fields and countries not represented in the sample. The type of data manipulation examined in this and previous related analyses is relatively unsophisticated. Future studies might need to assess whether other (and possibly subtler) forms of data misrepresentation are more or less common than this form, and how they might relate to various risk factors.

Second, our regression models, whilst including factors that are hypothesised to be relevant to scientific misconduct, may have excluded other relevant but unknown factors. Therefore, our regression estimates, like all regression estimates, are imperfect approximations and might still be biased due to omitted variables. These limitations entail that the specific numbers (odds ratios and probabilities) that we calculate should not be considered in absolute terms, but rather in relative terms, as theoretically meaningful comparisons between the relative importance of common risk factors. In other words, this study mainly assesses how factors make a greater or lesser difference *relative to each other*.

Third, our analysis focuses on data for first authors. The two main justifications for this choice are the fact that first authors are the main or sole beneficiaries of cash reward incentives [18], and the plausible assumption that first authors, who are typically the individuals who most contribute to the article, are most likely to be responsible for irregularities in the data. Our previous studies support the validity of this choice, by showing that predictors of misconduct tend to be more significantly associated with characteristics of first author relative to last author or an average of the team [6,14]. To further assess the validity of this assumption, we randomly sampled 20 articles and checked the contribution statements. In all 20 cases, the first author was indicated as having "performed the experiments", in 16 cases as also having analysed the data, and in most cases s/he was indicated as having contributed to all or most of the other roles. This strongly supports our assumption that the first author is most likely to be responsible for any manipulations in the data. However, absent a formal investigation, it is not possible to ascribe with certainty a suspected misconduct to a particular individual, and indeed in some cases more than one individual could have been involved. Further, it is conceivable that incentives for misconduct may differ depending on whether one is a trainee, a research scientist, a junior faculty member or a senior faculty member. Therefore, much of the complexity and diversity of phenomena underlying research misconduct may be obscured by imperfect measurement and heterogeneity in this analysis.

It should also be emphasized that, although our main results focused on forms of image duplication that are most likely to result from intentional manipulation, the specific nature and causes of these duplications was not ascertained, and a sizable proportion of them may still not constitute scientific misconduct defined as the intention to misrepresent research results. One cannot judge intent in figure construction from visual inspection alone, and categories 1–3 should be considered provisional binning for the severity of the types of problems found in figures rather than categories where the causes of the figure problems are known and understood. The exact proportion of data irregularities that are due to intentional manipulation is unknown and it is generally likely to vary by research area and journal. An in-depth investigation conducted at the journal *Molecular and Cell Biology* found that the vast majority of duplicated images were the result of honest error, and only 2/59 articles with image duplications (3.3%, 0.2% of all articles examined) were deemed to reflect an intention to deceive [30]. An analogous screening program at the *Journal of Clinical Investigation* found evidence of probable scientific misconduct in only 2/55 of images (3.6%, 1% of all articles examined) [31].

A few key strengths of this study should also be emphasised. First, the matched-control approach we adopted, combined with a conditional logistic regression, is an very powerful method, which allowed to control for crucial confounding factors that are rarely accounted for in studies on misconduct. Despite the non-random sampling, therefore, our results are quasi-experimental in nature. Second, this analysis employed a relatively large sample size. This gave it considerable statistical power: a post hoc analysis based on unconditional logistic regression suggested that the study had a statistical power of 99.9% to detect a small-size effect (i.e. OR = 1.5) when taking into account papers from all duplication categories (N = 879), and 99.7% when limited to categories 2 and 3 papers (N = 663). Therefore, whereas our results must be interpreted with caution with regards to their generalizability, they may be considered internally valid and highly precise comparisons between the relative importance of risk factors for scientific misconduct.

In light of their strengths and limitations, and combined with results of a previous analysis, these results suggest that scientific misconduct is a highly complex phenomenon, the nature and causes of which depend on interactions between multiple risk factors, operating at different levels. The probability to engage in misconduct may be highest amongst authors working in countries whose contribution to the international literature is rapidly expanding, such as

China and India, and particularly if they are rewarded with cash when they publish in international and high-impact journals. Within these countries, commonly hypothesised predictors of scientific misconduct, including being at an early-career stage and being incentivized by metrics of productivity and impact, may indeed be significant. However, in most other countries these factors may be irrelevant or have the opposite association with misconduct. Our results, like all observational studies, cannot conclusively prove causation, but they strongly suggest that the question of whether many common hypotheses and speculations about the "causes" of scientific misconduct are correct or not may depend on the socio-cultural context.

It is encouraging to observe that, whilst complex, these patterns follow a logic that can be partially understood, suggesting that the risk of misconduct and QRP may be reduced with appropriate preventive policies and initiatives. However, such policies and initiatives may need to be designed to address the specific needs and problems faced by a particular country or community.

## Supporting information

**S1 Data. csv file with all raw data, with UT codes anonymized.**
(CSV)

**S1 Code. R code with all analyses and secondary tests.**
(R)

## Acknowledgments

We thank Rodrigo Costas for contributing most of the individual-level bibliometric data. We thank Michael Kalichman, High Desmond and a third, anonymous reviewer for helpful comments. Hugh Desmond suggested the "shame" vs "guilt" culture hypothesis discussed in the text.

## Author Contributions

**Conceptualization:** Daniele Fanelli, Matteo Schleicher.

**Data curation:** Daniele Fanelli, Matteo Schleicher, Ferric C. Fang, Arturo Casadevall, Elisabeth M. Bik.

**Formal analysis:** Daniele Fanelli.

**Methodology:** Daniele Fanelli.

**Supervision:** Daniele Fanelli.

**Validation:** Daniele Fanelli.

**Visualization:** Daniele Fanelli.

**Writing – original draft:** Daniele Fanelli.

**Writing – review & editing:** Daniele Fanelli, Ferric C. Fang, Arturo Casadevall, Elisabeth M. Bik.

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
