## [Decision Letter · Decision Letter 0]

27 Apr 2021

PONE-D-20-40614

Individual and institutional predictors of misconduct are modulated by national publication incentives policy

PLOS ONE

Dear Dr. Fanelli,

Thank you for submitting your manuscript to PLOS ONE. After careful consideration, we feel that it has merit but does not fully meet PLOS ONE’s publication criteria as it currently stands. Therefore, we invite you to submit a revised version of the manuscript that addresses the points raised during the review process.

Your manuscript was carefully reviewed by experts in the field that made important comments and suggestions that if addressed  rigorously will greatly improve with interesting work.  Please ready carefully and address every comment and suggestion as best as you can. Make changes, in the manuscript, if you find appropriate. I am looking forward to receive the revised version at your earliest convenience. Thank you.

We look forward to receiving your revised manuscript.

Kind regards,

Cesario Bianchi

Academic Editor

PLOS ONE

Additional Editor Comments:

Dear Dr. Fanelli,

Your manuscript was reviewed by 3 experts that found your manuscript of interest. Scientific misconduct is a very important topic that has brought more attention with the high number of papers about covid-19 been retracted recently.

Please, ready carefully all comments and address thoroughly every single one. They are all very meaningful and answering them will greatly improve your interesting work. I personally would like to see some examples of image misconduct e correctness. I am also I little concerned about addressing misconduct by countries. I agree with one of the reviewers that would like to see some data from other countries ( is the reviewer example us turkey) and Brazil, for example.

More than once the reviewers would like more methodological details.

Cash incentives may come from direct payment per article but may be an important factor for fellowships and/or grant renewals.

Journal Requirements:

2. Please note that according to our submission guidelines (http://journals.plos.org/plosone/s/submission-guidelines), outmoded terms and potentially stigmatizing labels should be changed to more current, acceptable terminology. In order to avoid conflation between gender and sex, "female” or "male" should be changed to "woman” or "man" as appropriate.

We note that one or more of the authors are employed by a commercial company: Harbers Bik LLC.

3.1. Please provide an amended Funding Statement declaring this commercial affiliation, as well as a statement regarding the Role of Funders in your study. If the funding organization did not play a role in the study design, data collection and analysis, decision to publish, or preparation of the manuscript and only provided financial support in the form of authors' salaries and/or research materials, please review your statements relating to the author contributions, and ensure you have specifically and accurately indicated the role(s) that these authors had in your study. You can update author roles in the Author Contributions section of the online submission form.

3.2. Please also provide an updated Competing Interests Statement declaring this commercial affiliation along with any other relevant declarations relating to employment, consultancy, patents, products in development, or marketed products, etc.  

Reviewers' comments:

Reviewer's Responses to Questions

**Comments to the Author**

1. Is the manuscript technically sound, and do the data support the conclusions?

Reviewer #1: Yes

Reviewer #2: Yes

Reviewer #3: Partly

2. Has the statistical analysis been performed appropriately and rigorously? 

Reviewer #1: Yes

Reviewer #2: I Don't Know

Reviewer #3: Yes

3. Have the authors made all data underlying the findings in their manuscript fully available?

Reviewer #1: Yes

Reviewer #2: Yes

Reviewer #3: Yes

4. Is the manuscript presented in an intelligible fashion and written in standard English?

Reviewer #1: Yes

Reviewer #2: Yes

Reviewer #3: Yes

5. Review Comments to the Author

Reviewer #1: This is an interesting manuscript that re-uses an illustrative dataset in order to disentangle individual and country level predictors for image duplication in research articles. They an show that once country effects are controlled for some factors for which prior evidence is inconclusive do not appear to have a significant impact.

The analysis is based on a dataset that was used before. The description of the dataset used for the study at hand on p. 4 is somewhat confusing. Is the same sample as in ref [14] used here or a larger sample? Please also provide some basic information about the data and do not expect the reader to get into ref [14] and [15] to search for this information.

The analysis is based on a case-control matching. Unfortunately, the matching approach is described very briefly on p.6. It suggests that the country information is the only information used for the matching. But this seems unlikely. Hence, I would like to see some more information:

- Are the matched articles from the same year

- Same journal or at least subfield

The authors only match on the first affiliation mentioned on the article assuming that the lead author is the responsible author. Can the authors provide a robustness check for single-authored studies? If the sample becomes to small then I wonder whether the results hold if the authors match on the country combination of the author team.

Institution rankings vary over time. Would it be possible to use the time-varying ranking for your analysis?

Reviewer #2: A few general comments the authors may want to consider:

1. Although I understand the general principles behind the methods of analyses used, I don't have sufficient expertise to judge whether they have been performed appropriately and rigorously.

2. Isn’t it possible that differences in findings of research misconduct are based on factors other than whether the research misconduct occurred? For example, what if researchers in the US, Canada, and the UK have the resources, awareness, etc. to do a better job of cheating (e.g., using images from some other project so there are no duplicates in the same paper)?

3. The manuscript recognizes that duplication of images is only one form of misconduct. In theory, if the factors highlighted are correct (i.e., the importance of financial incentives), then this would generalize to many other forms of misrepresentation of research findings as well as simply sloppy research practices. That said, it seems duplicate images in a given paper seems like a very specific kind of misconduct, and one which is pretty unsophisticated (e.g., a moment’s thought would recognize that people often get caught for this misconduct, and being caught is easily avoided by just using other images that aren’t in the same paper). It may warrant some discussion that if inappropriate incentives are driving this kind of misconduct, then it is likely only the tip of the iceberg and/or something very different is going on here then with other kinds of misconduct.

4. If a paper has duplicate images, that could be something all authors were complicit in, but it’s at least as plausible that the duplication was committed by one of the authors and the other authors were insufficiently knowledgeable and/or attentive to have noticed the duplication. In that case, if a connection is being made to financial incentives for publication, it would be important to be clear (a) who gets the incentive (all authors, first author, corresponding author) and (b) who did what (e.g., if the paper has an author contributions section it might be possible to determine which author(s) was/were responsible for the images in question).

5. Finally, while I don’t think it’s essential for publication of this work, shouldn’t it be possible to run a fairly simple experiment of using the factors identified as predictors of image duplication by selecting 10 examples of duplication of images and 20 controls from each of 2 or 3 other journals?

Reviewer #3: Peer Review

Hugh Desmond

IHPST (CNRS/Paris I)

University of Antwerp

In this article, the authors re-analyze previous data on publications involving image manipulation, in order to gain a more precise understanding of the potential causes of patterns of misconduct. The wider significance of this research is to investigate the “misaligned incentive” view of scientific misconduct. According to this view, the prevalence of scientific misconduct is not simply due to isolated “bad apples” – ill-intentioned and/or anti-social individuals – but also due to the broader incentive system in science. The authors find that this view is only partially supported: only in some countries (i.e., China and South Korea, which are characterized by cash incentives) do (career) incentives lead to increased misconduct. This adds to a previous study by Fanelli et al. 2019.

This is a well-constructed study and a valuable contribution to the literature. The misaligned incentive view is indeed still quite prevalent – often it is even the default view. It’s not clear it deserves that dominance. So this study is a welcome contribution to the debate about the true causes of patterns of research misconduct.

My comments concern how the study is advertised. A safer but less interesting title would be: “Productive early career academics in low ranking institutions in South Korea and China more likely to duplicate images.” Justification is lacking why the results support conclusions about misconduct in general or about cash incentives in general (and not just cash incentives *in China*). Moreover, there are disclaimers cautioning the reader against reading too much into the study – which is of course fine, but puzzling nonetheless, since the authors’ disclaimers seem to to directly undercut the way the study is advertised.

Comment 1 -- Scope of conclusion: misconduct or image duplication?

This study is advertised as concerning “misconduct”, even though it only tests for factors that predict image duplication. However, do patterns of behavior regarding image duplication necessarily hold for patterns of misconduct more generally?

The disclaimer on lines 515ff (see comment 3) is especially puzzling, since the authors caution against drawing any conclusions about misconduct in general.

Moreover, there is a concrete reason why the null result found for Western researchers may not hold for misconduct generally. Image duplication is an egregious form of misconduct, and by 2013/2014 (the years of the analyzed studies) there would have been strong awareness of the norm of FFP for researchers in Western institutions. Moreover, image duplication is relatively easy to detect. So by this reasoning, the incentives to engage in egregious misconduct would not be strong enough: probability of being caught is too high, and researchers are aware of this probability. However, career incentives might still promote more subtle forms of misconduct. For instance: salami slicing, guest authorship, or bullying are all patterns of misconduct/QRP that are difficult to charge scientists with. In fact, in some contexts such as the USA, only FFP is categorized as misconduct. So, it’s not that misaligned incentives do not show up in misconduct, they just show up in different, less detectable forms of misconduct. (Compare the situation to financial incentives and sports doping.)

So, there should be additional argumentation why the study holds for misconduct in general, and more emphasis on just how the results are to be interpreted. Or else the scope of the title & abstract should be narrowed.

Comment 2 -- Role of cash incentives

The authors conclude that “Most of the differential effects observed in China and Korea relative to other countries may be plausibly explained as resulting directly or indirectly from the incentives created by a cash” (lines 468-470).

While I agree that it is “plausible”, given that this is the headline result of the article, the reader would expect additional justification. There are many other factors that could explain the differential effects observed in China and Korea.

a. Cash incentives vs. norms of academic hierarchy

The authors introduce the category of “german-developmental”, but don’t use it in the analysis and interpretation of the findings. Nonetheless, Figure 2c documents how in “german-developmental” countries, low institutional ranking of the first author predicts image duplication with statistical significance. So why could one not conclude that the increased image duplication in China/S-Korea is being caused by strict academic hierarchies – or not *also* being caused by this?

Obviously the authors are limited here by the available data. It would be nice to have data from, for instance, Turkey, to test whether we see the same incidence of data manipulation. In absence of such data, what can be inferred about the role of cash incentives *in particular*?

b. Cash incentives vs. RI culture

The role of RI culture is acknowledged in lines 102-109, but it does not shape the discussion of the findings. Nonetheless, by many accounts, China has only in the past few years become more serious about educating researchers on RI norms. So, given that the data is from 2013/14, the observed patterns might reflect the regulatory situation rather than cash incentives in particular. This would also imply that the pattern of image duplication might not hold as strongly – or not at all – in 2021.

To test this, one would need data from a scientifically active country with weak RI culture and with no cash incentives.

c. Cash incentives vs. non-WEIRD psychology

If you would introduce cash incentives in Germany, would image duplication shoot up dramatically? This is what the takeaway from this study implies. Yet somehow this scenario strikes seems implausible. There is much more at stake.

For instance, one could point to research on how moral psychology in e.g. China seems to work quite differently, with a much stronger emphasis on “shame” rather than “guilt” (cf. Henrich 2020). In a shame culture, there is a very strong incentive to meet social expectations -- whatever those might be, and regardless of the *means* by which one reaches these expectations. The moral unit is the collective, not the individual, and shame only arises from being caught (not from doing the deed). In this view, cash incentives only matter insofar they express social expectations and the value of the collective – and hence only secondarily are about cash. By this line of reasoning, the difference-maker is not cash-incentives policy, but rather shame vs. guilt cultures.

Comment 3 -- Conflicting signals

At the very end, there are some sentences that read like a disclaimer (lines 515-519). The authors caution the reader that the results may not apply to “other forms of misconduct” or to “countries not represented in the sample”. If this disclaimer is true, then shouldn’t the takeaway from this study be revised?

Minor Comments

Comment 4

Line 12 “A third improvement from [14] consists in testing three new variables that measure two commonly invoked risk factors.”

This wording might be a bit confusing for the reader – I found myself looking for the “three new variables”. Perhaps clarify that what follows are the two risk factors.

Comment 5

In the beginning the main result is described as: “the risk of problematic image duplication was higher for highly productive, ***highly cited***, early career researchers working in low-ranking institutions”

However, in Figure 2, citation rate is not seen to have any statistically significant relation to image duplication. Therefore, why is it mentioned as a result? Moreover, if it is not a statistically significant predictive factor, then why do a simulation of it at all (Figure 3)?

Comment 6

line 574-575 : “Therefore, we can conclude that many common hypotheses and speculations about the “causes” of scientific misconduct are likely to be correct or not, depending on the socio-cultural context.”

Sentence doesn’t seem quite right. I presume the authors mean this: “Therefore, we can conclude that THE QUESTION OF WHETHER many common hypotheses and speculations about the “causes” of scientific misconduct are correct or not, DEPENDS on the socio-cultural context.”

(Also, if this is the actual conclusion, why not have it reflected in title, abstract, introduction? Referring to the “socio-cultural context” is a more safer conclusion that referring to “cash incentive policies” in particular, and covers the roles academic hierarchy norms, RI cultures, or moral psychology might play.)

Comment 7

Line 576: “It is encouraging to observe that, whilst complex, these patterns follow a logic that can be understood, suggesting that the risk of misconduct and QRP may be reduced with appropriate preventive policies and initiatives.”

But what logic though? If upheld, the paper seems to contribute to undermining the dominant logic, namely the “misaligned incentive” view. If this view is false, then research misconduct may prove to be much more intractable (in Western countries) than previously hoped?

In fact, it would have been interesting to have heard slightly more about what the potential policy implications of this study would be. Should we give up on trying to nudge researcher behaviour through incentive engineering? (For a logic why incentive engineering cannot safeguard trust, see Desmond forthcoming in Philosophy of Science, http://philsci-archive.pitt.edu/18661/)

6. PLOS authors have the option to publish the peer review history of their article (what does this mean?). If published, this will include your full peer review and any attached files.

Reviewer #1: No

Reviewer #2: **Yes: **Michael Kalichman

Reviewer #3: **Yes: **Hugh Desmond

---

## [Author Response · Author response to Decision Letter 0]

28 May 2021

A rebuttal to each reviewer comment is included in the Pdf

---

## [Decision Letter · Decision Letter 1]

22 Oct 2021

PONE-D-20-40614R1Are individual and institutional predictors of misconduct modulated by national publication incentives policy? Results of a matched-control analysis of problematic image duplications.PLOS ONE

Dear Dr. Fanelli,

Thank you for patience whilst we completed further evaluation of your manuscript. Please accept our apologies once again for the inconvenience caused by the rescinding of your previous Accept decision.

After careful consideration, we feel that the manuscript has merit but does not fully meet PLOS ONE’s publication criteria as it currently stands. Your manuscript has been assessed by a member of PLOS ONE’s statistical advisory board (Reviewer 3) and I am pleased to let you know that no concerns have been raised as part of this review. However, following internal editorial review of the manuscript, there are a number of revisions to the text that you should perform before the manuscript can be accepted for publication. These are noted as follows:

Several concerns have been noted over the methodology for assigning gender:We note that you report a high percentage of unclassified authors from Chinese language names. Has the genderapi service been validated for names from all countries assessed in the manuscript? If not, this should be discussed as a potential limitation in the Discussion section.We understand that a gender was assigned to all unclassified names. Please provide additional details concerning how this manual gender assignment was performed, including specific protocols and a validation of this approach. Whilst we note that you have referred readers to Reference 14 for further information on this approach, this reference does not provide sufficient details for how this was done. This information should be included in the Methods section.  It is indicated that most unclassified names were Chinese. Please update your Methods section to provide the exact figure.

3.The article title implies conclusions/investigation of a causal relationship and frames the research around misconduct. We request a revision to the title for the following reasons:

a)The title suggests that predictors of misconduct are modulated by national publication incentives policies. However, as noted in your Discussion, your manuscript cannot prove causation. The title of the manuscript should be updated to clearly indicate this.

b)The use of “misconduct” in the title also requires additional nuance, since the manuscript does not directly measure misconduct (it only measures problematic image duplications)

c)We suggest the following revision:

A matched-control analysis examining what individual and institutional characteristics are associated with increased rates of problematic image duplications in scientific publications.

4. Please update Reference 14 to include the issue, page numbers and correct year, as it is currently marked as ‘in press’.

Figure 4 requires clarification: It is missing x-axis labels, and either (a) n values for each x-axis category should be specified in the legend, or (b) if “frequency” actually means “# of articles” then n values aren’t needed but the y-axis should be labelled more clearly.

6. We understand that you obtained ethical approval to conduct this study. Please update the Methods section of your manuscript indicate this, along with the name of the body that provided ethical approval for the study to take place.

7. We note that you have indicated that ‘All relevant data are within the manuscript and its Supporting Information files’. However, we note that not all of the analysis described in the manuscript can be reproduced from the data file provided. The datafile does not include the specific articles included in your analysis. We understand that you may not be able to make these data available with your manuscript. However, are you able to provide these data on request? If so, please include details of how such requests should be made in your Data Availability statement. Please note that it is not acceptable for authors to be the sole named individual responsible for ensuring data access. Your institution’s ethics or data protection committee regulate data access and act as a contact for requests of this data.

8. In addition, we would be grateful if you could provide an additional tab in your ‘S1_dataset.csv’ with a legend providing a definition of each of the columns.

We look forward to receiving your revised manuscript.

Kind regards,

Bradford Dubik, Associate Editor

On behalf of:

Cesario Bianchi

Academic Editor

PLOS ONE

Journal Requirements:

Reviewers' comments:

Reviewer's Responses to Questions

**Comments to the Author**

1. If the authors have adequately addressed your comments raised in a previous round of review and you feel that this manuscript is now acceptable for publication, you may indicate that here to bypass the “Comments to the Author” section, enter your conflict of interest statement in the “Confidential to Editor” section, and submit your "Accept" recommendation.

Reviewer #1: All comments have been addressed

Reviewer #3: All comments have been addressed

Reviewer #4: All comments have been addressed

2. Is the manuscript technically sound, and do the data support the conclusions?

Reviewer #1: Yes

Reviewer #3: Yes

Reviewer #4: Yes

3. Has the statistical analysis been performed appropriately and rigorously? 

Reviewer #1: Yes

Reviewer #3: Yes

Reviewer #4: Yes

4. Have the authors made all data underlying the findings in their manuscript fully available?

Reviewer #1: Yes

Reviewer #3: Yes

Reviewer #4: Yes

5. Is the manuscript presented in an intelligible fashion and written in standard English?

Reviewer #1: Yes

Reviewer #3: Yes

Reviewer #4: Yes

6. Review Comments to the Author

Reviewer #1: I am satisfied with the changes that have been made during the revision process.

To reach the character limit I add: congratulations!

Reviewer #3: (No Response)

Reviewer #4: (No Response)

7. PLOS authors have the option to publish the peer review history of their article (what does this mean?). If published, this will include your full peer review and any attached files.

Reviewer #1: No

Reviewer #3: **Yes: **Hugh Desmond

Reviewer #4: No

---

## [Author Response · Author response to Decision Letter 1]

16 Nov 2021

Thank you for your feedback, here below are responses to each query.

 1. Several concerns have been noted over the methodology for assigning gender:

 a. We note that you report a high percentage of unclassified authors from Chinese language names. Has the genderapi service been validated for names from all countries assessed in the manuscript? If not, this should be discussed as a potential limitation in the Discussion section.

 b. We understand that a gender was assigned to all unclassified names. Please provide additional details concerning how this manual gender assignment was performed, including specific protocols and a validation of this approach. Whilst we note that you have referred readers to Reference 14 for further information on this approach, this reference does not provide sufficient details for how this was done. This information should be included in the Methods section.  

RESPONSE: We agree that errors in gender classifications could affect the results, but we note a number of robustness checks that makes this unlikely. The percentage of unclassified author names was 5.8%, which is not really a "high" percentage. Of these, little over 50% were Chinese/Taiwanese, which is a relative majority but, again, not suggestive of a special imbalance.

As originally reported, the results were tested by re-assigning the unclassified names to either gender, and doing so makes not difference in the results. This would obviously be the case, since the percentage is small. 

More importantly, our conclusions about gender cut across countries, and are observed in US and Eu countries, where disambiguation errors are much rarer because names are typically gendered. 

 The Methods section was altered as follows:

"Most names were classified by a commercial service (genderapi.com), and the records that remained unclassified were subsequently completed by hand, via direct inspection of information available online. In particular, CVs and professional webpages were inspected for explicitly gendering information and, if this was not available, based on and photos, see [14] for details). This hand-coding was blind to the outcome (presence/absence of duplication in the paper). Names that remained unclassified (N=51/879, 5.8%) were re-coded as “male”. We used male as the default gender, based on the fact that most (i.e. 27) of these 51 unclassified names were Chinese or Taiwanese, and that about 75% of scientists in China are male [20]. The effect of this coding choice on results was checked with robustness tests reported in the Results section." 

The Discussion was extended with: 

"Precision in gender classification tends to be lower for East Asian countries, and a small percentage of names was not classifiable and coded as male (5.8%). However, classifying these names as female did not alter the results. More importantly, the lack of difference between genders was observed in Western-European and North-American countries, where names are typically gendered and therefore errors in gender attribution are extremely low. " 

 2. It is indicated that most unclassified names were Chinese. Please update your Methods section to provide the exact figure.

RESPONSE: The number is 27, which is a little above 50% the total. this detail was added in the methods section as shown above.

3.The article title implies conclusions/investigation of a causal relationship and frames the research around misconduct. We request a revision to the title for the following reasons:

a)The title suggests that predictors of misconduct are modulated by national publication incentives policies. However, as noted in your Discussion, your manuscript cannot prove causation. The title of the manuscript should be updated to clearly indicate this.

b)The use of “misconduct” in the title also requires additional nuance, since the manuscript does not directly measure misconduct (it only measures problematic image duplications)

c)We suggest the following revision:

 1. A matched-control analysis examining what individual and institutional characteristics are associated with increased rates of problematic image duplications in scientific publications.

RESPONSE: We appreciate the concerns about causal implications, and note that the choice of words like "predictor" in the title was indeed intended to avoid the concept of causation (predictor is a statistical term). It should also be noticed that the first part of the title is intended to describe the substantive hypotheses being tested, which are indeed causal hypotheses and are not our own hypotheses but hypotheses proposed in the literature. These hypotheses are about scientific misconduct in general.

 Our study is a test of such hypotheses done on image duplications and, whilst observational in nature, it does come close to a causal analysis by using a matched control, and it is intended to produce evidence about such theoretical hypotheses. 

All that said, we are happy to try to remove any ambiguity or strong causal implications in the title. We appreciate the suggestion made by the editors, but we feel that the title suggested doesn't quite capture what this study was about - rather, it describes the previous analysis of which this study was a follow-up.

We changed the title to: "Do individual and institutional predictors of misconduct vary by country? Results of a matched-control analysis of problematic image duplications."

4. Please update Reference 14 to include the issue, page numbers and correct year, as it is currently marked as ‘in press’.

RESPONSE: amended

 5. Figure 4 requires clarification: It is missing x-axis labels, and either (a) n values for each x-axis category should be specified in the legend, or (b) if “frequency” actually means “# of articles” then n values aren’t needed but the y-axis should be labelled more clearly.

RESPONSE: We appreciate the suggestion to improve clarity. The figure was corrected. 

Title now reads:

 "severity of image duplication, by country", y-axis is titled "n of articles" 

And the legend explains: 

"Number and percentage of problematic image duplications in our sample, partitioned by country of first author and by severity of duplication (where category 3 is the most problematic, because it is most likely to reflect intentional misrepresentation of data)."

6. We understand that you obtained ethical approval to conduct this study. Please update the Methods section of your manuscript indicate this, along with the name of the body that provided ethical approval for the study to take place.

RESPONSE: The methods section now includes the sentence:

"This project was reviewed and approved by the Research Ethics Committee of the London School of Economics and Political Science."

7. We note that you have indicated that ‘All relevant data are within the manuscript and its Supporting Information files’. However, we note that not all of the analysis described in the manuscript can be reproduced from the data file provided. The datafile does not include the specific articles included in your analysis. We understand that you may not be able to make these data available with your manuscript. However, are you able to provide these data on request? If so, please include details of how such requests should be made in your Data Availability statement. Please note that it is not acceptable for authors to be the sole named individual responsible for ensuring data access. Your institution’s ethics or data protection committee regulate data access and act as a contact for requests of this data.

RESPONSE: All identifying information from the original manuscripts is removed for ethical reasons, and therefore cannot be shared. However, we confirm that all the data required to reproduce all analyses in the study is available, albeit in an anonymised form.

8. In addition, we would be grateful if you could provide an additional tab in your ‘S1_dataset.csv’ with a legend providing a definition of each of the columns.

RESPONSE: A "Column legend" tab has been added to the file.

---

## [Editor Report · Decision Letter 2]

2 Feb 2022

Do individual and institutional predictors of misconduct vary by country? Results of a matched-control analysis o f problematic image duplications.

PONE-D-20-40614R2

Dear Dr. Fanelli,

We’re pleased to inform you that your manuscript has been judged scientifically suitable for publication and will be formally accepted for publication once it meets all outstanding technical requirements.

Kind regards,

George Vousden

Deputy Editor-in-Chief

PLOS ONE

Additional Editor Comments (optional):

Please note that the original Academic Editor handling your manuscript became unavailable and I have taken over its handling.
---

## [Editor Report · Acceptance letter]

28 Jul 2021

PONE-D-20-40614R1 

Are individual and institutional predictors of misconduct modulated by national publication incentives policy? Results of a matched-control analysis of problematic image duplications. 

Dear Dr. Fanelli:

I'm pleased to inform you that your manuscript has been deemed suitable for publication in PLOS ONE. Congratulations! Your manuscript is now with our production department. 

Kind regards, 

on behalf of

Dr. Cesario Bianchi 

Academic Editor

PLOS ONE